# Physiological responses of *Cucurbita pepo* seeds to cadmium and copper stress: Differential impacts on reserve mobilization, metabolic efficiency, and growth

Smail Acila[1,2]*, Nora Allioui[3], Samir Derouiche[2,4]

1 Department of Biology, Faculty of Nature and Life Sciences, University of El Oued, El Oued, Algeria,
2 Laboratory of Biodiversity and Application of Biotechnology in the Agricultural Field, University of El Oued, El Oued, Algeria, 3 Department of Ecology and Environmental Engineering, Faculty of Nature and Life Sciences and Earth and Universe Sciences, University of May 8th, 1945 Guelma, Guelma, Algeria, 4 Department of Cellular and Molecular Biology, Faculty of Nature and Life Sciences, University of El Oued, El Oued, Algeria

* smailacila@gmail.com, smail-acila@univ-eloued.dz

## Abstract

Heavy metal contamination poses a significant threat to agricultural productivity. This study investigated the physiological and biochemical responses of *Cucurbita pepo* seeds to cadmium (Cd) and copper (Cu) stress (100–200 µM) during germination. Although germination rates remained high (86.67–93.33%), seed vigor indices declined significantly under metal stress. Cadmium exhibited stronger growth inhibition, reducing total seedling length by 63.02% at 200 µM, whereas copper primarily affected biomass accumulation, reducing the seedling weight-based vigor index (SVI$_W$) by 40.4%. Biochemical analyses revealed metal-specific impacts on reserve mobilization. Cadmium exposure (200 µM) decreased soluble sugars in cotyledons by 16%, while maintaining protein content at 106% of control levels, indicating inhibition of protein degradation and impaired reserve utilization. In contrast, copper at 100 µM increased cotyledonary sugars by 63%, reflecting its dual role as both a micronutrient and stressor. Principal component analysis confirmed the greater toxicity of Cd, which explained 79.7% of the variance in metabolic disruption. These findings demonstrate that cadmium consistently impairs seedling establishment by disrupting nutrient mobilization pathways, while copper exhibits concentration-dependent effects, being stimulatory at low concentrations but inhibitory at higher levels. This study provides crucial insights into heavy metal phytotoxicity mechanisms and underscores the importance of monitoring metal pollution in agricultural systems to enhance crop resilience.

**Data availability statement:** All relevant data are within the manuscript and its figures files.

**Funding:** The author(s) received no specific funding for this work.

**Competing interests:** The authors have declared that no competing interests exist.

## Introduction

Heavy metal contamination represents an escalating global concern due to its profound impact on environmental and agricultural systems. The expansion of industrial activities, such as mining and urban transportation, has led to the widespread dissemination of heavy metals, including cadmium (Cd) and copper (Cu), into ecosystems. Unlike organic pollutants, heavy metals are non-biodegradable and persist in the environment, undergoing transformations that can enhance their bioavailability and toxicity [1]. This persistence results in the degradation of ecosystems quality, reduces agricultural productivity, and has adverse effects on human health [2].

While trace amounts of metals like Cu are essential for plant metabolic processes, their excessive accumulation becomes a toxic stressor, interfering with physiological pathways and leading to oxidative stress [3,4]. For instance, high concentrations of Cu can disrupt root growth and protein structures [ 5,6]. Cd, on the other hand, is inherently toxic even at low concentrations, inducing symptoms like inhibited photosynthesis, reduced nutrient absorption, and stunted plant growth [7,8]. Plants respond to heavy metal stress by activating enzymatic (e.g., catalase, superoxide dismutase) and non-enzymatic (e.g., glutathione, vitamin C) antioxidant defense mechanisms, which mitigate oxidative damage [9].

Heavy metals stress also disrupts the mobilization of seed reserves during germination, a critical phase for plant development. Studies suggest that metals interfere with enzymatic activities necessary for the breakdown and transport of storage molecules like proteins, starch, and lipids to the embryonic tissues [10]. This disruption leads to impaired seedling growth and reduced metabolic efficiency, further compounding the adverse effects on plant development [11].

Zucchini (*Cucurbita pepo L.*) is an economically significant crop cultivated worldwide for its nutritional and medicinal value. It is a rich source of potassium, beta-carotene, proteins, and bioactive compounds such as phenolics and flavonoids [12,13]. However, its early developmental stages, such as seed germination and seedling growth, are particularly sensitive to environmental stresses, including heavy metals exposure [14]. These stages lack robust defense mechanisms, making them highly vulnerable to disruptions in metabolic and physiological processes caused by heavy metals [15]. Moreover, heavy metal stress not only affects seed germination rates but also impacts subsequent growth by altering cellular structures, reducing photosynthetic activity, and impairing nutrient uptake [16].

Despite extensive research on heavy metal toxicity in various crops, limited studies have specifically examined the differential impacts of Cd and Cu on reserve mobilization efficiency during early seedling development in *Cucurbita pepo*. Furthermore, the comparative analysis of metabolic efficiency parameters and multivariate relationships between physiological and biochemical responses under controlled metal stress conditions remains underexplored.

Building upon our previous investigation of oxidative stress responses in embryonic axes under identical Cd and Cu exposure conditions [15], the current study provides a complementary analysis focused specifically on reserve mobilization dynamics. We examine how heavy metals affect the utilization of seed reserves from

cotyledons and their allocation to embryonic axes during germination. This integrated approach addresses the identified knowledge gap and offers a more comprehensive understanding of heavy metal phytotoxicity mechanisms in zucchini seeds.

## Materials and methods

### Seed preparation protocol

Zucchini (*Cucurbita pepo* L.) seeds of the Quarantaine variety, from the same batch used in our previous study [15], were processed using established protocols. Seeds were thoroughly cleaned with tap water to remove thiram residues, then disinfected in 10% *(v/v)* sodium hypochlorite solution (prepared from commercial NaClO containing 12° chlorometric degree) for 10 minutes. After extensive rinsing with distilled water, seeds were submerged in distilled water at 5°C for 30 minutes to synchronize germination, following our previously validated method [15].

### Germination assay and metal exposure

Post-soaking, randomly selected zucchini seeds were placed in sterilized glass Petri dishes (15 cm diameter × 2 cm height) lined with filter paper, with each Petri dish containing 50 seeds. Treatments included cadmium chloride ($CdCl_2$) and copper chloride ($CuCl_2$) solutions. Seeds were exposed daily to either distilled water (control) or solutions of $CdCl_2$ and $CuCl_2$ at 100 µM and 200 µM concentrations, with 15 ml of the respective solution applied per Petri dish. The solutions were completely replaced daily to maintain consistent metal concentrations throughout the 8-day germination period. Each treatment was replicated three times (three independent Petri dishes per treatment). The germination assay was conducted in an incubator maintained at 27 ± 1°C in complete darkness for 8 days (Fig 1).

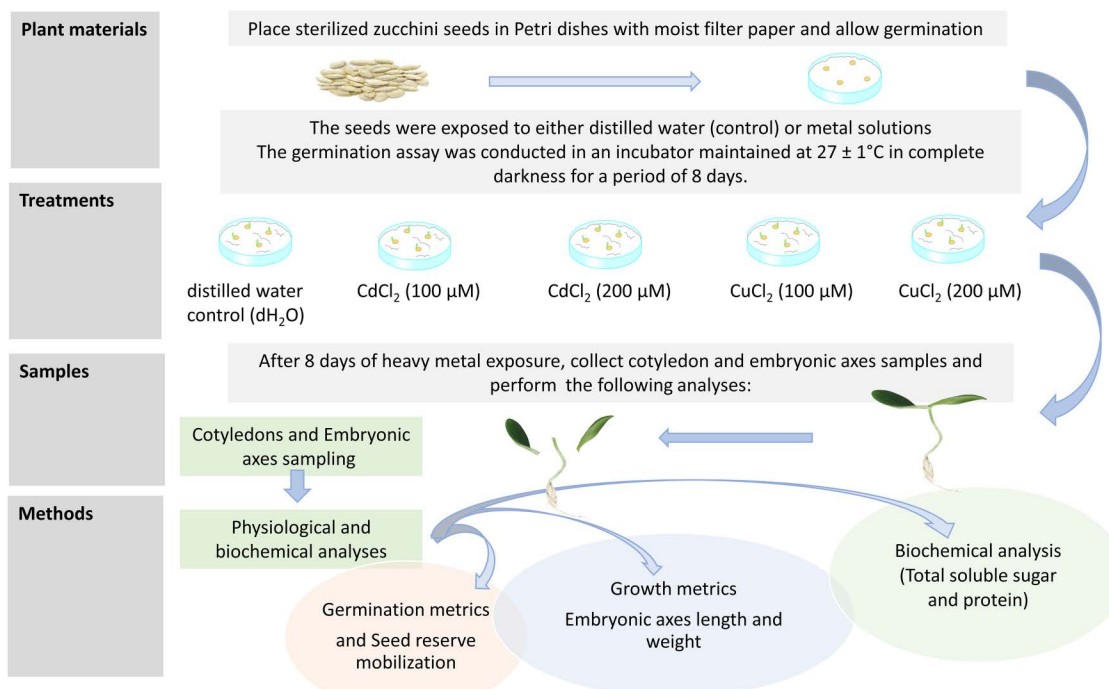

**Fig 1. Schematic representation of the experimental setup comparing Cd and Cu-stressed *Cucurbita pepo* seedlings with unstressed controls.**

The selection of metal concentrations (100–200 μM) was based on preliminary tests showing significant physiological responses in our experimental system [15]. These concentrations are relevant to environments impacted by pollution, as previous studies have shown that cadmium, lead, and copper can reach elevated levels in contaminated agricultural soils due to industrial activities and improper waste management practices [17,18].

## Assessment parameters and measurement protocols

### Germination metrics.

1. **Germination Percentage (GP):** This metric represents the maximum proportion of seeds that germinated within the 8 days, expressed as a percentage of the total seeds tested. A seed was considered germinated when its radicle reached a minimum length of 2 mm.

2. **Seed vigor index based on seedling length ($SVI_L$) and Seed vigor index based on seedling weight ($SVI_W$):** Calculated after the 8-day germination period using the formula proposed by [19]:

$$SVI_L = (GP \times TSL) / 100$$

$$SVI_W = (GP \times TSW) / 100$$

where: TSL: Total seedling length (mm); TSW: Total seedling weight (mg); GP: Germination percentage (%)

**Seed reserve mobilization and conversion efficiency analysis.** The mobilization and conversion efficiency of seed reserves in *C. pepo* were evaluated through multiple parameters following established protocols. All measurements were performed on dry weight basis after oven-drying samples at 80 °C for 48 h until constant weight was achieved.

### 1. Depleted seed reserve weight (SRW)

The weight of depleted seed reserves (SRW) was determined following the methodology described by Pereira, Pereira [20], using the following equation:

$$SRW \ (mg) = SDWi - CDWr$$

where SDWi represents the initial seed dry weight and CDWr denotes the residual dry weight of cotyledons.

### 2. Seed reserve depletion rate (SRR)

The SRR parameter quantifies the depletion rate of stored seed dry matter and was calculated following Pereira, Pereira [20]:

$$SRR \ (\%) = (SRW/SDWi) \times 100$$

### 3. Seed metabolic efficiency (SME)

The concept of seed metabolic efficiency (SME), as introduced by Rao and Sinha [21], represents the ratio of the hypocotyl and radicle dry weight (g) to the depleted seed dry weight (g). A higher SME value reflects reduced seed efficiency since more seed reserves are consumed in the development of radicle and hypocotyl.

The amount of seed material utilized through respiration (SMR) was determined using the equation of Rao and Sinha [21]:

$$SMR = SDWi - (HW + RW + CDWr)$$

where: SDWi = Seed dry weight before germination; HW = Hypocotyl dry weight; RW = Radicle dry weight; CDWr = Remaining cotyledon dry weight

The seed metabolic efficiency (SME) was then computed using the formula provided by Rao and Sinha [21]:

$$SME = HW + RW/ SMR$$

### 4. Energy efficiency (EE)

According to Andrade, Coelho [22], EE was calculated using the following formula:

$$EE (\%) = ((SDWi - CDWr - TSW) / SDWi) \times 100$$

**Growth measurements of *C. pepo* seedlings.** For each of the three biological replicates per treatment, 20 randomly selected 8-day-old zucchini seedlings were measured. The length (in cm, using millimeter paper) and dry weight (in mg, after oven-drying at 60°C for 48 h) of the radicle, hypocotyl, and total seedling were recorded. Values from the 20 seedlings were averaged to obtain a single mean value per replicate for statistical analysis.

**Heavy metals phytotoxicity assessment:** The impact of heavy metals on embryonic axis (EA) growth was quantified using the formulas described by Wierzbicka, Bemowska-Kałabun [23]:

$$EA \text{ Length Phytotoxicity } (\%) = ((EAL_{control} - EAL_{treatment}) / EAL_{control}) \times 100$$

$$EA \text{ Weight Phytotoxicity } (\%) = ((EAW_{control} - EAW_{treatment}) / EAW_{control}) \times 100$$

**Biochemical content of Zucchini seedling.** *Sample processing for biochemical analyses:* For biochemical analyses, composite samples were prepared from pooled tissues to obtain sufficient material. For cotyledon analyses (sugars and proteins), tissues from 20 seedlings per biological replicate were pooled. For embryonic axis sugar content, tissues from 30 seedlings per replicate were pooled due to the small size of individual axes. All samples were freeze-dried (lyophilization) to preserve metabolic integrity and ground to a fine powder using a mortar and pestle before extraction.

### 1. Total soluble sugar content of cotyledons

*Sample preparation and extraction*

Freeze-dried cotyledon powder (0.5 g) was mixed with 10 mL of 80% (v/v) methanol in a sealed glass vial. The homogenized mixture was stored in darkness for 48 hours at 4°C with occasional shaking. The filtrate was evaporated under a gentle stream of nitrogen until completely dry. The methanol extraction method was selected for its efficiency in extracting soluble sugars while minimizing interference from other compounds [24,25].

*Total soluble sugar assay*

Following DuBois, Gilles [26] protocol, 30 mg of dry extract was mixed with 1 ml methanol and 4 ml distilled water. 50 µl of this aqueous extract was combined with 3 ml sulfuric acid and 1 ml phenol solution (5%). Samples were heated in a water bath at 100°C for 3 minutes, cooled, and absorbance was measured at 490 nm using a UV-Vis spectrophotometer (Shimadzu UV-1800). A glucose standard curve was prepared using concentrations ranging from 0.16 to 1.78 mg/ml, yielding a linear calibration with $r^2 = 0.9708$. Results were expressed as µg glucose equivalent per mg dry weight (µg GE/mg DW).

## 2.  Protein quantification in cotyledons

*Sample preparation*

Freeze-dried cotyledon powder (20 mg) was mixed with 5 ml of 2% NaOH to obtain a protein extract (4 mg/ml). The mixture was incubated at 60°C for 30 minutes to enhance protein solubilization, then</mark> centrifuged for 10 minutes at 5000 × g.

*Protein assay*

Protein quantification was performed according to the Bradford method [27]. 200 µl of supernatant was mixed with 2 ml of Coomassie Blue reagent in darkness. Absorbance was measured at 595 nm using the same spectrophotometer.

A standard curve was prepared using bovine serum albumin (BSA) concentrations ranging from 0.001 to 0.008 mg/ml, yielding a linear calibration with $r^2 = 0.9886$. Results were expressed as µg BSA equivalent per mg dry weight (µg BSA/mg DW).

## 3.  Total soluble sugar content of embryonic axes

Embryonic axes were carefully excised from seedlings of each treatment group, including controls. The pooled samples were processed as described for cotyledons (freeze-drying and grinding). The same protocol utilized for quantifying sugars in cotyledons was applied for embryonic axes, and results were similarly expressed as µg glucose equivalent per mg dry weight (µg GE/mg DW).

## Statistical analysis

All experiments were conducted with three independent biological replicates (n = 3), where each replicate consisted of a Petri dish containing 50 seeds. For growth parameters (lengths and dry weights), 20 seedlings were randomly selected from each replicate, and their values were averaged to obtain a single mean value per biological replicate prior to statistical analysis. This approach ensured that the Petri dish was maintained as the experimental unit, thus avoiding pseudoreplication.

Data were expressed as means ± standard error (SE) of the three biological replicates. Statistical analysis was conducted using Minitab (version 16). Differences among treatments were tested for significance using Fisher's Least Significant Difference (LSD) test at a probability level of $p < 0.05$. Principal Component Analysis (PCA) was performed with XLSTAT (version 16) using the replicate means to explore the multivariate relationships among the measured physiological and biochemical parameters.

## Results

### Germination performance of *C. pepo* seeds under cadmium and copper stress

**Germination percentage (GP).**  The 8-day germination study of *Cucurbita pepo* seeds revealed intriguing results regarding their response to metal exposure (Table 1). In the control group, where seeds were germinated in distilled water, a high GP (93.33%) was observed. When subjected to specific concentrations of 100 µM and 200 µM cadmium chloride ($CdCl_2$) and copper chloride ($CuCl_2$) solutions, the seeds exhibited only a minor, statistically insignificant (p = 0.313) decrease in average GP. Notably, even at a relatively high concentration of 200 µM $CdCl_2$, the seeds maintained a GP of 86.67%, which did not differ significantly from the control. Similarly, exposure to 100 µM $CuCl_2$ resulted in a minimal reduction of just 1.43%, with a GP of 92%.

**Seed vigor index based on seedling weight ($SVI_W$).**  The seed vigor index based on seedling weight ($SVI_W$) showed significant differences among treatments (Table 1). The control group ($dH_2O$) had the highest $SVI_W$ value of 2127.1,

**Table 1. Germination percentage (GP) and seed vigor indexes ($SVI_L$ and $SVI_W$) of zucchini seeds (*C. pepo* L.) after 8 days of germination under $CdCl_2$ and $CuCl_2$ stress. dH$_2$O: distilled water.**

| Metal Solution | Treatment | GP (%) | Inhibition (%) | $SVI_L$ | Inhibition (%) | $SVI_W$ | Inhibition (%) |
|---|---|---|---|---|---|---|---|
| Control | dH$_2$O | 93.33±0.67[a] | – | 1092.5±7.80[a] | – | 2127.1±15.2[a] | – |
| $CdCl_2$ | 100 µM | 88.0±1.15[a] | (5.71) | 607.20±7.97[c] | (44.42) | 1524.2±20.0[b] | (28.35) |
| | 200 µM | 86.67±4.37[a] | (7.14) | 382.6±19.3[e] | (64.98) | 1413.5±71.3[b] | (33.55) |
| $CuCl_2$ | 100 µM | 92.0±2.0[a] | (1.43) | 680.8±14.8[b] | (37.68) | 1438.0±31.3[b] | (32.40) |
| | 200 µM | 89.33±1.76[a] | (4.28) | 541.8±10.7[d] | (50.41) | 1268.5±25.0[c] | (40.36) |
| P. value | | 0.313[NS] | – | 0.000[***] | – | 0.000[***] | – |

Data are presented as means±SE (n=3 independent experiments). Means not sharing the same letter are significantly different at p≤0.05 by Fisher's LSD test. [NS] not significant, [***] Extremely significant.

significantly outperforming all metal-treated groups (p<0.05). Among the metal treatments, seeds exposed to $CdCl_2$ at 100 µM exhibited an $SVI_W$ of 1524.2, corresponding to an inhibition rate of 28.3%. Similarly, $CuCl_2$ at 100 µM resulted in an $SVI_W$ of 1438.0 with an inhibition rate of 32.4%. At a higher concentration of 200 µM, $CdCl_2$ reduced the $SVI_W$ to 1413.5, with an inhibition rate of 33.5%, while $CuCl_2$ caused the most significant reduction, with an $SVI_W$ of 1268.5 and an inhibition rate of 40.4%.

**Seed vigor index based on seedling length ($SVI_L$).** A similar trend was observed for the seed vigor index based on seedling length ($SVI_L$). The control group (dH$_2$O) displayed the highest $SVI_L$ value of 1092.5 (Table 1), which was significantly greater than all metals-treated groups (p<0.05). Among the treatments, seeds exposed to $CuCl_2$ at 100 µM exhibited an $SVI_L$ of 680.8, with an inhibition rate of 37.7%, whereas $CdCl_2$ at the same concentration resulted in an $SVI_L$ of 607.2, with an inhibition rate of 44.4%. At a higher concentration of 200 µM, $CuCl_2$ reduced the $SVI_L$ to 541.8, corresponding to an inhibition rate of 50.4%, while $CdCl_2$ caused the most significant reduction in $SVI_L$, with a value of 382.6 and an inhibition rate of 65.0%.

## Reserve mobilization and conversion efficiency of *C. pepo* seeds under cadmium and copper stress

**Depleted seed reserve weight (SRW).** Analysis of variance revealed a highly significant effect (p<0.001) of metals treatments on the weight of depleted seed reserves (SRW) during germination. As shown in Fig 2, SRW decreased significantly under cadmium treatment, with values of 39.47 mg, and 33.83 mg at 100 µM and 200 µM, respectively, and under copper treatment, with values of 44.98 mg and 37.34 mg at 100 µM and 200 µM, respectively, compared to the control (58.07 mg). Metal stress was associated with reduced mobilization and utilization of seed nutrient reserves during germination.

**Seed reserve depletion rate (SRR).** Similarly, to SRW, SRR was significantly affected by metals (p<0.001), with decreases under cadmium at 100 µM (33.96%) and 200 µM (29.11%), and under copper at 100 µM (38.7%) and 200 µM (32.13%), compared to the control (49.97%).

**Seed material respired (SMR).** Analysis of variance showed a highly significant effect of metals treatments on the quantity of seed material respired (SMR) (p<0.001). According to Fig 2, SMR decreased significantly under cadmium, with values of 22.14 mg at 100 µM and 17.51 mg at 200 µM, and under copper, with values of 29.34 mg at 100 µM and 23.14 mg at 200 µM, compared to the control (35.28 mg). This reduction in SMR corresponded with lower reserves degradation, suggesting a potential disruption in seed respiratory metabolism.

**Seed metabolic efficiency (SME).** Metals treatments significantly affected seed metabolic efficiency (p<0.05). As shown in Fig 2, SME tended to increase under cadmium, reaching 1.244 at 200 µM, compared to the control (0.7053). In contrast, SME remained relatively similar to the control under copper treatment at 100 µM (0.7191) and 200 µM (1.017), with no significant variation.

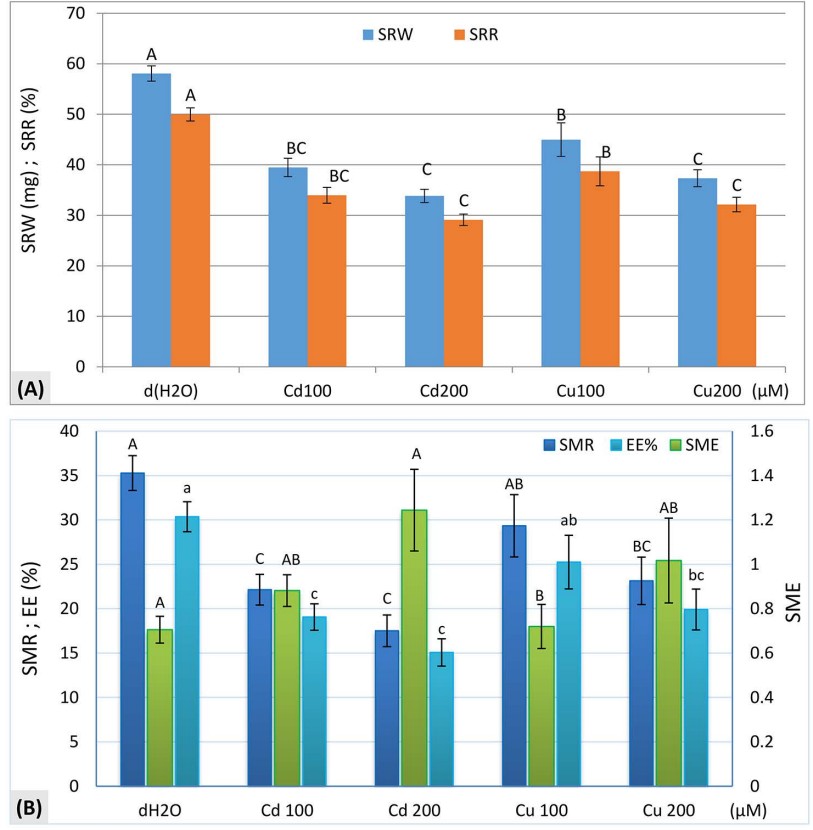

**Fig 2. Seed reserve mobilization and conversion efficiency in zucchini (*C. pepo* L.) seeds after 8 days of germination under CdCl₂ and CuCl₂ stress.** (A) Depleted seed reserve weight (SRW) and seed reserve depletion rate (SRR). (B) Seed metabolic efficiency (SME), seed material respired (SMR), and energy efficiency (EE). Values for the same parameter that do not share the same letter are significantly different at p ≤ 0.05, as determined by Fisher's LSD test.

**Energy depletion rate (EE).** Similar to SMR, metals treatments significantly (p < 0.001) affected EE (Fig 2), with decreases under cadmium at 100 µM (19.06%) and 200 µM (15.07%) and under copper at 100 µM (25.25%) and 200 µM (19.91%), compared to the control (30.36%). The most pronounced reduction in energy consumption occurred under 200 µM cadmium.

## Metal phytotoxicity on seedling growth metrics

**Metal phytotoxicity on seedling length metrics.** *Metal Phytotoxicity on Hypocotyl Length (HLP)*: Metals treatments significantly (p < 0.001) reduced hypocotyl length (Table 2). Cadmium at 100 µM and 200 µM reduced *HLP* by 45.08% and 58.60%, respectively. Copper reduced *HLP* by 45.38% at 100 µM and 47.81% at 200 µM. Cadmium showing stronger effects, particularly at 200 µM (Fig 3A).

*Metal Phytotoxicity on Radicle Length (RLP)*: Metals had a highly significant (p < 0.001) phytotoxic effect on radicle length (Table 2). Cadmium was more toxic, reducing *RLP* by 41.78% at 100 µM and 69.69% at 200 µM. Copper also showed a phytotoxic effect, with reductions of 26.48% and 51.80% at 100 µM and 200 µM, respectively. Once again, cadmium exhibited stronger, dose-dependent toxicity (Fig 3A).

*Metal Phytotoxicity on Total Seedling Length (TSLP)*: A highly significant (p < 0.001) effect of metals on total seedling length was observed (Table 2). Cadmium reduced *TSLP* by 43.98% at 100 µM and 63.02% at 200 µM, while copper reduced it by 38.93% and 49.63% at 100 µM and 200 µM, respectively. Similar to hypocotyl and radicle length, cadmium showed the most marked phytotoxicity, particularly at 200 µM (Fig 3A).

**Table 2.** Total seedling length (TSL), hypocotyl length (HL), radicle length (RL), HL/RL ratio, total seedling dry weight (TSW), hypocotyl dry weight (HW), radicle dry weight (RW), and HW/RW ratio in zucchini seedlings (*C. pepo* L.) after 8 days of germination under $CdCl_2$ and $CuCl_2$ stress. $dH_2O$: distilled water.

| Metal Solution | Treatment | HL (cm) | RL (cm) | HL/RL | TSL (cm) | TSW (mg) | HW/RW | RW (mg) | HW (mg) |
|---|---|---|---|---|---|---|---|---|---|
| Control | $dH_2O$ | $7.26\pm0.41^a$ | $4.45\pm0.28^a$ | $1.87\pm0.22^b$ | $11.71\pm0.31^a$ | $22.79\pm1.05^a$ | $2.49\pm0.23$ | $7.09\pm0.57^a$ | $15.71\pm0.73^a$ |
| $CdCl_2$ | 100 µM | $4.16\pm0.24^b$ | $2.75\pm0.20^{bc}$ | $1.77\pm0.26^b$ | $6.90\pm0.33^{bc}$ | $17.32\pm0.53^b$ | $2.95\pm0.20$ | $4.64\pm0.31^b$ | $12.68\pm0.39^b$ |
| | 200 µM | $3.02\pm0.18^c$ | $1.34\pm0.23^d$ | $3.29\pm0.42^a$ | $4.42\pm0.27^d$ | $16.32\pm0.62^{bc}$ | $3.91\pm0.24$ | $3.39\pm0.13^c$ | $12.93\pm0.58^b$ |
| $CuCl_2$ | 100 µM | $4.04\pm0.18^b$ | $3.36\pm0.38^b$ | $1.58\pm0.25^b$ | $7.40\pm0.40^b$ | $15.63\pm0.67^{bc}$ | $2.92\pm0.23$ | $4.18\pm0.25^{bc}$ | $11.45\pm0.59^{bc}$ |
| | 200 µM | $3.92\pm0.13^b$ | $2.15\pm0.16^c$ | $2.09\pm0.19^b$ | $6.07\pm0.21^c$ | $14.20\pm1.02^c$ | $2.55\pm0.16$ | $3.94\pm0.17^{bc}$ | $10.26\pm0.88^c$ |
| P. value | | 0.000*** | 0.000*** | 0.000*** | 0.000*** | 0.000*** | 0.000*** | 0.000*** | 0.000*** |

Data are presented as the means $\pm$ SE (n = 20). Values in the same column not sharing the same letter are significantly different at $p \leq 0.05$ by Fisher's LSD test. *** Extremely significant.

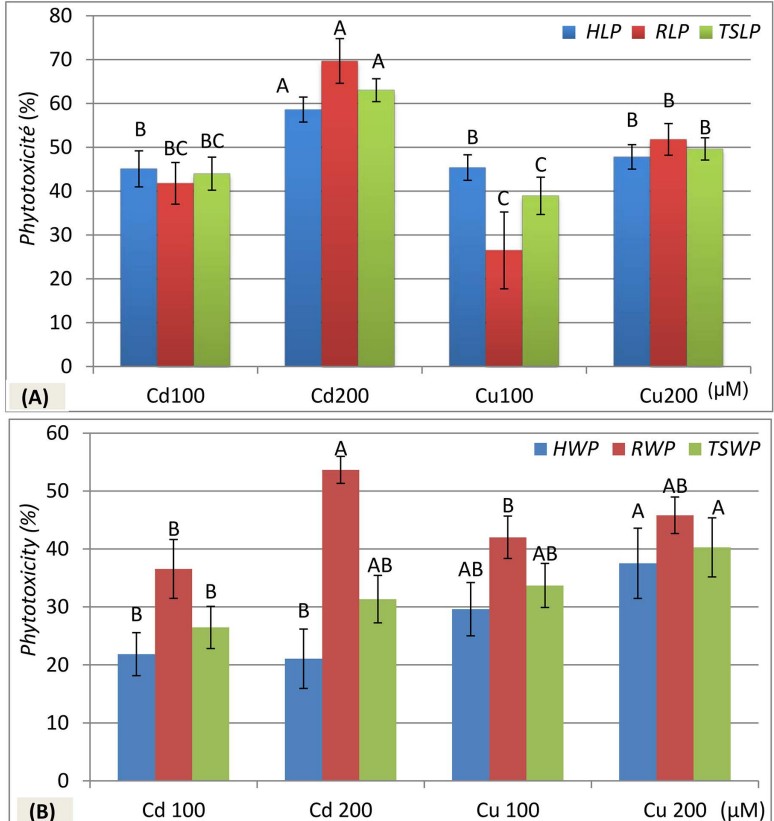

**Fig 3. Phytotoxicity of $CdCl_2$ and $CuCl_2$ stress on (A) length [hypocotyl (*HLP*), radicle (*RLP*), and total seedling (*TSLP*)] and (B) dry weight [hypocotyl (*HWP*), radicle (*RWP*), and total seedling (*TSWP*)] in zucchini (*C. pepo* L.) seedlings after 8 days of germination.** Values for the same parameter that do not share the same letter are significantly different at $p \leq 0.05$ (Fisher's LSD test). $dH_2O$: distilled water.

**Metal phytotoxicity on seedling dry weight metrics.** *Metal Phytotoxicity on Hypocotyl Dry Weight (HWP)*: Analysis of variance showed a significant effect (p < 0.001) of metals on hypocotyl dry weight (Table 2). Interestingly, copper treatment tended to increase *HWP*, reaching 29.62% at 100 µM and 37.54% at 200 µM, compared to cadmium, which resulted in lower *HWP* values of 21.86% at 100 µM and 21.07% at 200 µM (Fig 3B).

*Metal Phytotoxicity on Radicle Dry Weight (RWP)*: Results showed a significant effect (p < 0.001) of metals on radicle dry weight (Table 2). As illustrated in Fig 3B, cadmium caused pronounced phytotoxic effects, with *RWP* increasing by 36.57% at 100 µM and 53.66% at 200 µM. Copper also increased *RWP*, with values rising by 42.03% and 45.83% at 100 µM and 200 µM, respectively. Cadmium demonstrated greater toxicity than copper, particularly at higher concentrations.

*Metal Phytotoxicity on Total Seedling Dry Weight (TSWP)*: Metals significantly (p < 0.001) affected the total seedling dry weight, as shown in Table 2. Fig 3B illustrates that *TSWP* increased under copper treatment by 33.72% at 100 µM and 40.29% at 200 µM compared to cadmium, which caused increases of 26.47% at 100 µM and 31.36% at 200 µM.

The phytotoxic effects of Cd and Cu on seedling morphology were visually evident, with pronounced stunting and reduced biomass accumulation under heavy metal concentrations (Fig 4).

## Biochemical content of Zucchini seedling

**Total soluble sugars content in cotyledons.** One-way ANOVA revealed a significant effect (p < 0.001) of metals treatments on the total soluble sugars content in cotyledons (Fig 5A). Cadmium at 100 µM and 200 µM reduced sugars content to 13.090 µg GE/mg DW (−7.6% of control) and 11.902 µg GE/mg DW (−16% of control), respectively, compared to the control at 14.177 µg GE/mg DW. Conversely, copper at 100 µM significantly increased sugars content to 23.164 µg GE/mg DW (+ 63% of control), but at 200 µM, the sugars content returned to a level (13.22 µg GE/mg DW) not significantly different from the control.

**Protein content in cotyledons.** One-way ANOVA indicated a significant effect (p < 0.01) of metal treatments on protein content in cotyledons (Fig 5B). After eight days of zucchini seed germination, protein content in cotyledons was significantly higher compared to the control (1.328 mg BSA/g DW). Under cadmium treatment, protein content remained elevated at 1.771 mg BSA/g DW (+33%) at 100 µM and 2.733 mg BSA/g DW (+106%) at 200 µM compared to the control. Similarly, under copper treatment, protein content recorded 1.686 mg BSA/g DW (+27%) at 100 µM and 1.909 mg BSA/g DW (+44%) at 200 µM.

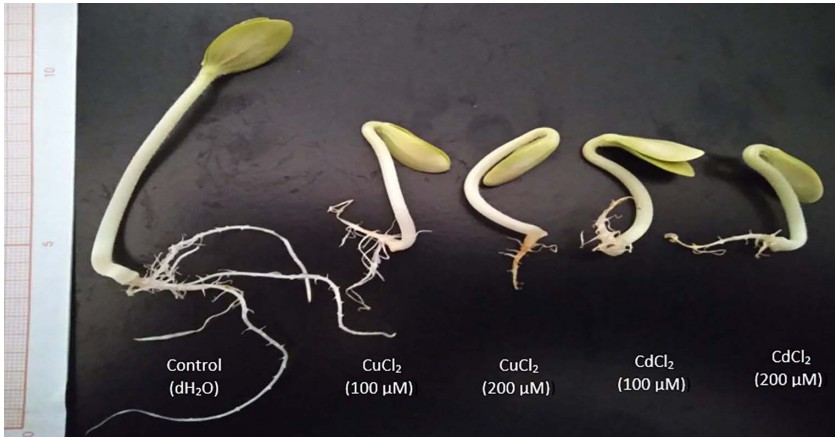

**Fig 4. Visual phenotypic responses of zucchini (*C. pepo*) seedlings to CdCl$_2$ and CuCl$_2$ exposure.**

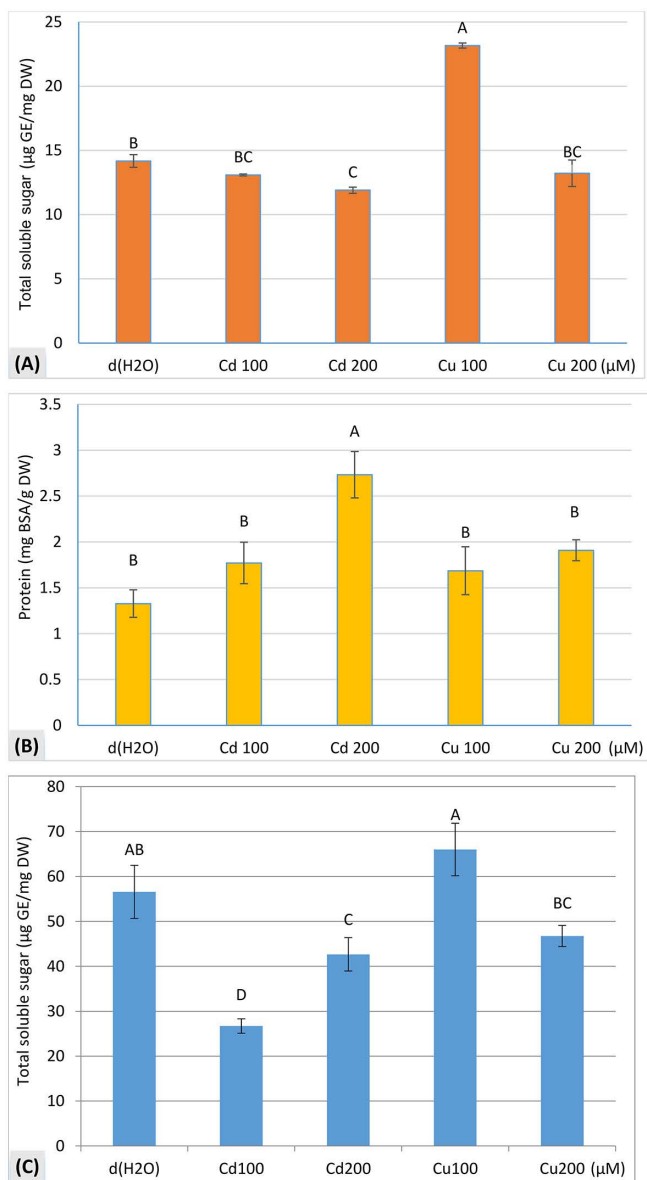

**Fig 5. Biochemical content of zucchini (*C. pepo* L.) seedlings after 8 days of germination under CdCl₂ and CuCl₂ stress. (A)** Total soluble sugars content of cotyledons, **(B)** Protein content of cotyledons, and **(C)** Total soluble sugars content of embryonic axes. Values for the same parameter that are labeled with different letters are significantly different at $p \leq 0.05$, as determined by Fisher's LSD test.

**Total soluble sugars content in embryonic axes.** One-way ANOVA revealed a highly significant effect ($p < 0.001$) of metals treatments on the total soluble sugars content in the embryonic axes of zucchini seedlings (Fig 5C). Copper at 100 µM markedly increased sugars content to 66.01 µg GE/mg DW (+ 16.7%) compared to control (56.58 µg GE/mg DW). However, at 200 µM, copper reduced sugars content by 17.4% compared to the control. In contrast, cadmium drastically reduced sugars content, with decreases of 52.8% (26.70 µg GE/mg DW) at 100 µM and 24.6% (42.63 µg GE/mg DW) at 200 µM.

**Multivariate analysis of metal stress responses.** Principal Component Analysis (PCA) revealed two major dimensions (Fig 6) explaining 94.44% of total variance in the dataset (F1 = 79.73%, F2 = 14.72%). The first component (F1) represented a generalized stress response axis, with strong negative loadings from growth parameters (SVI$_L$: −0.997,

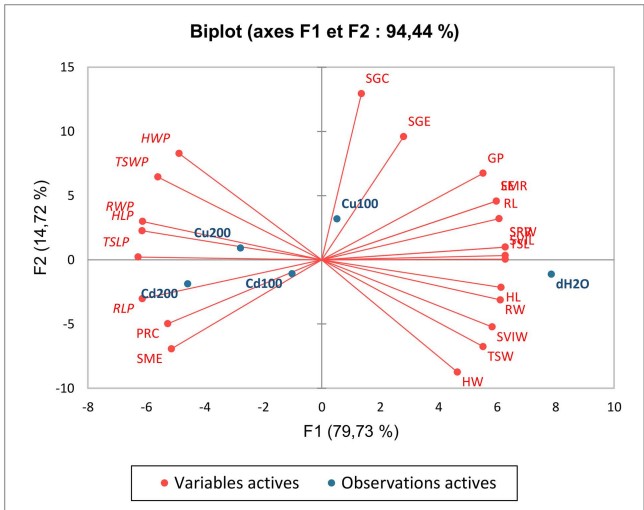

**Fig 6. Principal Component Analysis (PCA) biplot of physiological and biochemical responses in *Cucurbita pepo* under heavy metal stress, showing relationships among measured parameters along two principal components.** Factor 1 (horizontal axis, 79.7% variance) represents the primary stress response gradient, clearly separating growth-related variables (germination percentage [GP], seed vigor indices [SVI$_L$, SVI$_W$], seedling lengths [TSL, HL, RL], and seedling weights [TSW, HW, RW]) from phytotoxicity indicators (length and weight phytotoxicity measures [*TSLP, HLP, RLP, TSWP, HWP, RWP*]). Factor 2 (vertical axis, 14.7% variance) reveals an independent metabolic adaptation axis, distinguishing biochemical parameters (seed reserve metrics [SRW, SRR, SMR], metabolic efficiency [SME, EE], and biochemical contents [SGC, PRC, SGE]) from morphological traits.

TSL: −0.995, HW: −0.974) versus positive loadings from phytotoxicity indices (*RLP*: 0.973, *HLP*: 0.974). This clear separation confirmed the dose-dependent inhibition pattern observed in univariate analyses.

The second component (F2) reflected metabolic adaptation strategies, showing positive associations with biochemical markers (SGC: 0.882, SGE: 0.653) contrasting with negative loadings from morphological traits (HW: −0.596, SVI$_W$: −0.357). Notably, protein content (PRC) showed divergent behavior with moderate loadings on both axes (F1: −0.835, F2: −0.339), suggesting complex regulation under metal stress.

Pearson correlation matrix (Table 3) revealed three distinct clusters of interrelated variables: (1) growth parameters (germination percentage [GP], seed vigor indices [SVI$_L$, SVI$_W$], seedling lengths [TSL, HL, RL], and seedling weights [TSW, HW, RW]) showing strong positive correlations among themselves (r = 0.88–0.99) and negative correlations with (2) phytotoxicity indicators (length and weight phytotoxicity measures [*TSLP, HLP, RLP, TSWP, HWP, RWP*]; r = −0.77 to −0.99). A third cluster of (3) biochemical parameters (seed reserve metrics [SRW, SRR, SMR], metabolic efficiency [SME, EE], and biochemical contents [SGC, PRC, SGE]) demonstrates variable association patterns, with metabolic efficiency measures showing inverse relationships with several growth parameters (r = −0.81 to −0.89).

The matrix highlights several key relationships: (a) germination percentage maintains strong positive correlations with all growth measures (r > 0.87) and negative correlations with phytotoxicity markers (r < −0.80); (b) seed reserve depletion (SRR) shows moderate positive associations with seedling vigor indices (r = 0.62–0.68); and (c) biochemical contents exhibit differential correlation patterns, with soluble sugars in embryonic axes (SGE) showing positive association with metabolic efficiency (r = 0.57 with SME) but negative correlation with phytotoxicity (r = −0.51 with RLP). These correlation patterns quantitatively support the physiological relationships observed in experimental treatments and revealed through PCA.

## Discussion

The study revealed significant insights into the physiological responses of *Cucurbita pepo* seeds to cadmium (Cd) and copper (Cu) stress during germination. Heavy metals exposure is well-documented to impair plant development through

**Table 3. Correlation matrix of physiological and biochemical parameters in *Cucurbita pepo* under cadmium and copper stress, showing pairwise Pearson correlation coefficients (r values).**

| Variable | GP | SVI$_L$ | SVI$_W$ | SRW | SRR | SMR | SME | EE | TSL | HL | RL | TSLP | HLP | RLP | TSW | HW | RW | TSWP | HWP | RWP | PRC | SGC | SGE |
|---|---|---|---|---|---|---|---|---|---|---|---|---|---|---|---|---|---|---|---|---|---|---|---|
| GP | 1 | | | | | | | | | | | | | | | | | | | | | | |
| SVI$_L$ | **0.884** | 1 | | | | | | | | | | | | | | | | | | | | | |
| SVI$_W$ | 0.650 | **0.905** | 1 | | | | | | | | | | | | | | | | | | | | |
| SRW | **0.910** | **0.991** | **0.904** | 1 | | | | | | | | | | | | | | | | | | | |
| SRR | **0.910** | **0.991** | **0.904** | **1.000** | 1 | | | | | | | | | | | | | | | | | | |
| SMR | **0.984** | **0.953** | 0.766 | **0.967** | **0.967** | 1 | | | | | | | | | | | | | | | | | |
| SME | **−0.881** | −0.824 | −0.574 | −0.825 | −0.825 | **−0.895** | 1 | | | | | | | | | | | | | | | | |
| EE | **0.984** | **0.953** | 0.766 | **0.967** | **0.967** | **1.000** | −0.895 | 1 | | | | | | | | | | | | | | | |
| TSL | 0.871 | **0.999** | **0.908** | **0.987** | **0.987** | **0.944** | −0.822 | **0.944** | 1 | | | | | | | | | | | | | | |
| HL | 0.797 | **0.980** | **0.936** | **0.957** | **0.957** | **0.884** | −0.704 | 0.885 | **0.982** | 1 | | | | | | | | | | | | | |
| RL | **0.920** | **0.965** | 0.808 | **0.964** | **0.964** | **0.968** | −0.942 | 0.969 | 0.963 | 0.896 | 1 | | | | | | | | | | | | |
| TSLP | −0.869 | **−0.999** | **−0.917** | **−0.988** | **−0.988** | **−0.942** | 0.801 | −0.942 | −0.999 | −0.987 | −0.954 | 1 | | | | | | | | | | | |
| HLP | −0.798 | **−0.978** | **−0.944** | **−0.960** | **−0.960** | **−0.884** | 0.691 | −0.884 | −0.979 | −0.999 | −0.889 | **0.986** | 1 | | | | | | | | | | |
| RLP | **−0.932** | **−0.975** | −0.823 | **−0.977** | **−0.977** | **−0.979** | 0.925 | −0.979 | −0.973 | −0.912 | −0.998 | 0.966 | 0.908 | 1 | | | | | | | | | |
| TSW | 0.555 | 0.851 | **0.993** | 0.847 | 0.847 | 0.684 | −0.495 | 0.684 | 0.857 | **0.898** | 0.742 | −0.867 | −0.906 | −0.756 | 1 | | | | | | | | |
| HW | 0.372 | 0.695 | **0.934** | 0.706 | 0.707 | 0.508 | −0.306 | 0.509 | 0.701 | 0.755 | 0.573 | −0.715 | −0.770 | −0.588 | **0.965** | 1 | | | | | | | |
| RW | 0.747 | **0.970** | **0.957** | **0.944** | **0.944** | 0.851 | −0.702 | 0.851 | **0.976** | **0.992** | **0.892** | **−0.978** | **−0.990** | −0.903 | **0.929** | 0.801 | 1 | | | | | | |
| TSWP | −0.577 | −0.866 | **−0.996** | −0.861 | −0.861 | −0.704 | 0.512 | −0.704 | −0.871 | **−0.910** | −0.757 | **0.881** | 0.918 | 0.772 | **−1.000** | **−0.958** | **−0.939** | 1 | | | | | |
| HWP | −0.420 | −0.736 | **−0.953** | −0.745 | −0.745 | −0.555 | 0.351 | −0.555 | −0.741 | −0.792 | −0.616 | 0.755 | 0.806 | 0.631 | **−0.979** | **−0.998** | −0.834 | **0.973** | 1 | | | | |
| RWP | −0.757 | **−0.973** | **−0.958** | **−0.949** | **−0.949** | −0.859 | 0.702 | −0.859 | **−0.978** | **−0.994** | **−0.894** | **0.981** | **0.993** | **0.906** | **−0.929** | −0.801 | **−1.000** | **0.939** | 0.834 | 1 | | | |
| PRC | −0.846 | −0.865 | −0.604 | −0.826 | −0.826 | **−0.880** | **0.943** | −0.880 | −0.869 | −0.795 | −0.926 | 0.850 | 0.775 | **0.915** | −0.530 | −0.304 | −0.786 | 0.549 | 0.355 | 0.783 | 1 | | |
| SGC | 0.573 | 0.203 | −0.069 | 0.283 | 0.283 | 0.464 | −0.622 | 0.464 | 0.183 | 0.008 | 0.415 | −0.165 | −0.010 | −0.399 | −0.159 | −0.241 | −0.025 | 0.145 | 0.218 | 0.015 | −0.380 | 1 | |
| SGE | 0.766 | 0.434 | 0.231 | 0.526 | 0.526 | 0.658 | −0.486 | 0.658 | 0.404 | 0.325 | 0.483 | −0.415 | −0.345 | −0.511 | 0.131 | 0.045 | 0.238 | −0.152 | −0.076 | −0.258 | −0.339 | 0.735 | 1 |

The bold values are significantly different from 0 at a significance level of alpha = 0.05.

oxidative stress, disruption of enzymatic activity, and nutrient imbalance [28]. Consistent with these findings, our results demonstrate a dose-dependent inhibition of germination percentage, embryonic axis growth, and seed reserve mobilization under both Cd and Cu stress.

## Seed germination responses to cadmium and copper stress

Heavy metals exposure is known to induce oxidative stress, disrupt enzymatic activity, and create nutrient imbalances, which are critical factors affecting seed germination and seedling growth. Specifically, cadmium has been shown to significantly impair germination rates and seedling vigor by inducing oxidative stress, which leads to the production of reactive oxygen species (ROS) that damage cellular components [29,30]. However, at moderate concentrations (100–200 µM), the protective role of seed coat barriers may have mitigated these effects, as observed in our study.

Our findings reveal that *Cucurbita pepo* seed germination was not significantly inhibited by cadmium chloride ($CdCl_2$) or copper chloride ($CuCl_2$) at concentrations of 100 and 200 µM, as the emergence of a 2 mm radicle—a key marker of germination—occurred even in heavy metal-stressed media. This aligns with previous research [31], suggesting that the seed's structural defenses play a vital role in early-stage germination. Its coat, composed of lignin, cellulose, hemicellulose, and pectin, likely acts as a physical and chemical barrier, limiting the mobility of metal ions and immobilizing them at the cell wall level [32,33]. Such protective mechanisms are consistent with findings by Wierzbicka and Obidzińska [34], who reported similar effects across various plant species.

However, under higher concentrations of Cd and Cu, germination percentage (GP) and seed vigor indexes ($SVI_L$ and $SVI_W$) were significantly suppressed. This reduction is linked to the toxicity of these metals, which interfere with critical metabolic processes rather than water uptake [35]. The more severe inhibition observed under Cd stress compared to Cu stress is consistent with its known ability to disrupt membrane integrity and cellular respiration [36].

## Impacts of cadmium and copper stress on reserve mobilization and conversion efficiency in *C. pepo* seeds

The mobilization and conversion of seed reserves are critical for seedling establishment, and both processes were significantly impacted under cadmium (Cd) and copper (Cu) stress. Our findings demonstrate that reserve mobilization efficiency, as indicated by reduced seed reserve depletion (SRW, SRR) and, was hindered by both metals.

The observed disruptions in soluble sugar dynamics and sustained protein levels in cotyledons under metal stress suggest potential interference with enzymatic processes involved in reserve mobilization. While the current study did not directly measure hydrolytic enzyme activities such as α-amylase or proteases, the metabolic patterns observed align with established mechanisms of metal-induced enzyme inhibition [37,38].

The maintained protein levels in cotyledons under Cd stress may indicate impaired proteolytic activity, possibly through direct metal binding to sulfhydryl groups in protease active sites [38]. However, these interpretations remain inferential, and direct enzyme activity measurements would be required for definitive mechanistic conclusions.

The reductions in SRW and SRR were more pronounced under Cd stress, suggesting a stronger inhibitory effect on reserve mobilization compared to Cu. This aligns with Cd's higher affinity for binding sulfhydryl groups in proteins and enzymes, which could potentially disrupt critical enzymatic activities involved in the breakdown of starches and proteins [38].

Interestingly, low concentrations of Cu (100 µM) appeared to enhance certain metabolic responses, likely due to adaptive metabolic responses as suggested in our previous work [15]. However, at higher concentrations (200 µM), both Cd and Cu reduced the efficiency of nutrient mobilization, as reflected by declines in reserve depletion metrics (SRW and SRR).

This inhibition was accompanied by a reductions in the amount of material respired (SMR) and seed metabolic efficiency (SME), highlighting disruptions in energy metabolism—a phenomenon consistent with findings from previous studies [39].

## Impacts of heavy metals on embryonic development and seedling growth

The inhibition of key metabolic pathways due to heavy metals stress can lead to reduced growth and development, as seen in various studies on different plant species, including *Cucurbita pepo* [40,41]. Research indicates that copper toxicity significantly affects germination rates and carbohydrate metabolism, which is essential for energy supply during early growth stages [42]. Our results corroborate these findings, as seedling growth under Cu stress exhibited a marked reduction in vigor and metabolic efficiency.

Heavy metal stress, particularly at concentrations of 200 µM, significantly inhibited seedling growth, as evidenced by reduced radicle and hypocotyl elongation. This growth suppression could be attributed to metal ion penetration into embryonic tissues, which may cause cellular toxicity and disrupt metabolic processes critical for development [43].

Notably, our complementary study on the same plant material [15] revealed significant alterations in antioxidant enzyme activities (SOD, GST) in embryonic axes under identical metal exposure conditions, demonstrating the capacity of heavy metals to modulate enzymatic systems in germinating zucchini seeds.

At lower concentrations, zucchini seedlings displayed adaptive responses, including increased dry weight and improved tolerance. These responses may be attributed to structural adaptations, such as sclerophylly (as evidenced by increased tissue density in seedlings under similar metal stress [44]), cell wall reinforcement, and regulated membrane permeability, which mitigate the effects of metal toxicity [45,46]. Our complementary study on seedlings further supports this, showing that *Cucurbita pepo* activates physiological defenses like sclerophylly and carotenoid synthesis to cope with Cd and Cu toxicity [44]. Notably, the toxicity of copper (Cu) surpassed that of cadmium (Cd) at equivalent concentrations, consistent with findings by Ghori, Ghori [16].

Cd exhibited a more pronounced phytotoxic effect on the embryonic axis, significantly reducing its length and weight. This aligns with studies indicating that Cd interferes with auxin signaling pathways and disrupts microtubule organization, thereby inhibiting cell elongation and division [23]. In contrast, Cu's deleterious effects, while substantial, were less severe, potentially due to its essential role in enzymatic functions and redox balance in trace amounts [6].

A critical factor limiting seedling establishment under heavy metals stress was the impaired mobilization of seed reserves. Disruptions in amino acid release and soluble sugars transport from cotyledons to the embryo under Cd and Cu stress highlight the central role of nutrient mobilization in early seedling development [7,47]. These findings underscore the differential impacts of Cd and Cu on seed germination and growth, with Cd exerting a more severe inhibitory effect, particularly on embryonic axis development and metabolic efficiency.

## Biochemical responses to heavy metal stress

Exposure to cadmium (Cd) and copper (Cu) significantly influenced soluble sugars and protein dynamics in zucchini seedlings. Soluble sugars levels in cotyledons declined under Cd stress in a dose-dependent manner, while low concentrations of Cu (100 µM) temporarily increased sugars content. The observed reduction in sugar mobilization at higher metal concentrations may reflect impaired nutrient mobilization essential for seedling growth [42]. The increase in soluble sugars in embryonic axes under Cd exposure (100–200 µM) is consistent with stress-induced accumulation of compatible solutes [48].

Protein content in cotyledons remained significantly higher under both Cd and Cu stress compared to the control, suggesting inhibition of protein degradation and disrupted reserve metabolism. This pattern is consistent with reports that high Cu concentrations can inhibit the breakdown of storage proteins [49], which could reduce amino acid availability for embryonic development. The sustained protein levels under Cd stress align with established mechanisms of protease inhibition through metal binding to sulfhydryl groups [38]. This likely contributes to impaired protein degradation, potentially limiting amino acid supply for embryonic growth.

Complementing these findings, our previous investigation on the same plant material under identical exposure conditions revealed that Cd and Cu also significantly alter antioxidant enzyme activities and oxidative stress markers in embryonic axes [15]. Together, these studies suggest that heavy metals disrupt multiple metabolic pathways during early seedling development.

The broader implications of these findings are evident across plant species; such as in *Vigna radiata*, Cd exposure has been shown to suppress protease activity [38], while in *Triticum aestivum*, both metals reduced aminopeptidase and endoprotease activity [50]. Research on *Zea mays* further indicates that Cd can disrupt metalloproteases [49].

The differential toxicity between Cd and Cu may stem from their distinct biochemical interactions: Cd's stronger thiol-group binding could potentially inhibit enzymes more severely and contribute to ROS generation, exacerbating oxidative stress [51]. Unlike Cu – which maintains essential roles at low concentrations – Cd forms stable cellular complexes that cause greater metabolic disruption [52,53], which could explain the more pronounced reductions in seedling vigor and metabolic efficiency observed under Cd stress.

Consistent with these biochemical alterations, phenotypic alterations (Fig 4) correlate with the quantitative reductions in hypocotyl length (Table 2) and the impaired reserve mobilization evidenced by sustained protein levels in cotyledons (Fig 2), supporting Cd's superior toxicity in this experimental system.

The multivariate PCA analysis provided a higher-order synthesis of these complex responses, revealing two distinct but complementary response axes (Fig 6). Factor 1 (79.7% variance) captured the core stress gradient, with phytotoxicity markers (*TSLP*, *RLP*) opposing growth parameters (SVI$_L$, TSL), quantitatively confirming Cd's greater inhibitory potency than Cu. Factor 2 (14.7%) revealed an independent metabolic adaptation axis, where sugar dynamics (SGC/SGE) dissociated from structural growth – explaining why low Cu concentrations stimulated sugar accumulation despite growth suppression. This multivariate pattern mirrors recent findings in metal-stressed legumes [47] and suggests a hierarchical response: biochemical adjustments (F2) precede morphological damage (F1).

These PCA-driven insights fundamentally explain how Cd and Cu interfere with reserve mobilization. While both metals disrupted enzymatic breakdown (as seen in F1's strong loading of SME [−0.815]), their differential positioning along F2 (Cu-associated metabolic plasticity vs. Cd's consistent toxicity) underscores their distinct biochemical modes of action during early development. Critically, the inhibited transport of reserves from cotyledons to embryonic axes – evident in F1's opposition between SRR and phytotoxicity markers – directly limited nutrient supply without affecting water uptake, ultimately hampering seedling establishment.

The Pearson correlation matrix further elucidated the physiological relationships observed under metal stress, revealing three key clusters of variables: (1) growth parameters, (2) phytotoxicity indicators, and (3) biochemical/metabolic markers. The strong positive correlations among growth parameters (e.g., SVI$_L$ and TSL; r > 0.88) and their negative correlations with phytotoxicity measures (e.g., *RLP* and *HLP*; r < −0.77) corroborate the dose-dependent growth inhibition patterns demonstrated in our experimental results. Notably, the inverse relationships between metabolic efficiency (SME) and seedling vigor (r = −0.81 to −0.89) suggest that metal stress disrupts the conversion of seed reserves into biomass, consistent with the observed accumulation of undegraded proteins in cotyledons (Fig 5B). These findings align with the PCA results, where F1 (79.7% variance) captured the growth-phytotoxicity gradient, while F2 (14.7%) reflected metabolic adaptation strategies. The differential correlation patterns of biochemical markers (e.g., SGE's positive association with SME but negative correlation with *RLP*) highlight the complex interplay between nutrient mobilization and metal toxicity, underscoring Cd's greater disruptive potency compared to Cu.

## Study limitations and future directions

While this study provides comprehensive insights into reserve mobilization under heavy metal stress, certain limitations should be acknowledged. First, the internal concentrations of Cd and Cu in different seed tissues were not directly measured. Future studies incorporating ICP-MS analysis would provide definitive evidence regarding metal uptake and distribution patterns. Second, specific enzymatic activities involved in reserve mobilization (e.g., α-amylase, proteases) were not quantified, limiting mechanistic interpretations to inferential levels based on physiological responses.

However, it is important to note that our parallel investigation on the same plant material under identical metal exposure conditions [15] provided complementary data on oxidative stress responses and antioxidant enzyme activities, offering a

more comprehensive understanding of the plant's defense mechanisms. The current study's focus on reserve mobilization efficiency, combined with our previous work on oxidative stress [15], provides an integrated perspective on heavy metal phytotoxicity in germinating zucchini seeds.

Additional limitations include the controlled laboratory conditions and the focus on early germination stages. Future research should explore field conditions and later developmental stages to assess agricultural relevance.

## Conclusion

This study demonstrates the physiological responses of *Cucurbita pepo* seeds to cadmium (Cd) and copper (Cu) stress during germination. While germination rates remained largely unaffected, significant physiological and biochemical disruptions were observed, particularly under higher concentrations of Cd and Cu (200 µM).

The seed coat likely limited metal ion penetration, but elevated metal levels impaired reserve mobilization and seedling growth, potentially through inhibition of key enzymatic activities such as α-amylase and acid phosphatase. Cd exhibited greater toxicity than Cu, as evidenced by sustained protein accumulation in cotyledons and impaired mobilization, though low Cu concentrations (100 µM) showed a stimulatory effect on metabolic activities, highlighting its dual role as both a micronutrient and a toxicant.

These findings underscore the complex adaptations of *Cucurbita pepo* to heavy metals stress, providing insights into their potential resilience mechanisms. Further research into molecular and antioxidant pathways could enhance phytoremediation strategies and improve crop performance in contaminated soils.

## Acknowledgments

The authors extend their gratitude to the Faculty of Nature and Life Sciences at the University of El-Oued for providing the laboratory facilities necessary to conduct this research.

## Author contributions

**Conceptualization:** Smail Acila, Nora Allioui, Samir Derouiche.

**Data curation:** Smail Acila, Nora Allioui, Samir Derouiche.

**Formal analysis:** Smail Acila.

**Investigation:** Smail Acila.

**Methodology:** Smail Acila.

**Project administration:** Smail Acila.

**Resources:** Smail Acila.

**Software:** Smail Acila.

**Supervision:** Smail Acila.

**Validation:** Smail Acila, Nora Allioui, Samir Derouiche.

**Visualization:** Smail Acila.

**Writing – original draft:** Smail Acila.

**Writing – review & editing:** Smail Acila, Nora Allioui, Samir Derouiche.

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
