## [Decision Letter · Decision Letter 0]

13 Sep 2025

Dear Dr. ACILA,

Thank you for submitting your manuscript to PLOS ONE. After careful consideration, we feel that it has merit but does not fully meet PLOS ONE’s publication criteria as it currently stands. Therefore, we invite you to submit a revised version of the manuscript that addresses the points raised during the review process.

https://journals.plos.org/plosone/s/submission-guidelines#loc-laboratory-protocols . Additionally, PLOS ONE offers an option for publishing peer-reviewed Lab Protocol articles, which describe protocols hosted on protocols.io. Read more information on sharing protocols at https://plos.org/protocols?utm_medium=editorial-email&utm_source=authorletters&utm_campaign=protocols .

We look forward to receiving your revised manuscript.

Kind regards,

Debasis Mitra

Academic Editor

PLOS ONE

**Journal Requirements:**

1. When submitting your revision, we need you to address these additional requirements. Please ensure that your manuscript meets PLOS ONE's style requirements, including those for file naming. The PLOS ONE style templates can be found at https://journals.plos.org/plosone/s/file?id=wjVg/PLOSOne_formatting_sample_main_body.pdf and https://journals.plos.org/plosone/s/file?id=ba62/PLOSOne_formatting_sample_title_authors_affiliations.pdf 2. Thank you for stating the following in the Acknowledgments Section of your manuscript: This research did not receive any specific grants; it is supported by the SNV Faculty, University of El-Oued (Algeria), and the Directorate-General for Scientific Research and Technological Development (DGRSDT), Ministry of Higher Education and Scientific Research (Algeria). We note that you have provided funding information that is not currently declared in your Funding Statement. However, funding information should not appear in the Acknowledgments section or other areas of your manuscript. We will only publish funding information present in the Funding Statement section of the online submission form. Please remove any funding-related text from the manuscript and let us know how you would like to update your Funding Statement. Currently, your Funding Statement reads as follows: The author(s) received no specific funding for this work.  Please include your amended statements within your cover letter; we will change the online submission form on your behalf. 3. Please include captions for your Supporting Information files at the end of your manuscript, and update any in-text citations to match accordingly. Please see our Supporting Information guidelines for more information: http://journals.plos.org/plosone/s/supporting-information. 4. If the reviewer comments include a recommendation to cite specific previously published works, please review and evaluate these publications to determine whether they are relevant and should be cited. There is no requirement to cite these works unless the editor has indicated otherwise. 

Reviewers' comments:

**Comments to the Author**

1. Is the manuscript technically sound, and do the data support the conclusions?

Reviewer #1: Yes

Reviewer #2: Yes

2. Has the statistical analysis been performed appropriately and rigorously?

Reviewer #1: Yes

Reviewer #2: Yes

3. Have the authors made all data underlying the findings in their manuscript fully available?

Reviewer #1: Yes

Reviewer #2: Yes

4. Is the manuscript presented in an intelligible fashion and written in standard English?

Reviewer #1: Yes

Reviewer #2: Yes

**Reviewer #1:**  I appreciate the chance to study the physiological and biochemical impacts of copper and cadmium stress on Cucurbita pepo seed germination. A significant topic of heavy metal contamination and its effects on crop development is covered in the study. Understanding seed reserve mobilization and seedling growth under metal stress is made possible by the experimental design and analyses.

I do have some recommendations, though, to enhance the manuscript's presentation, rigor, and clarity:

1. Introduction:

An explicit statement of the precise gap in the body of research that this study seeks to address would be beneficial, even though the introduction gives a good overview of heavy metal toxicity in plants. Justifying the study's importance will be easier if it is made clear how this research expands on existing knowledge. Streamlining these portions can improve readability and scientific formality. Additionally, some lines are redundant or written in a colloquial tone.

2. Discussion:

The results are interpreted in detail and connected to earlier research in the discussion, which enhances the text. There isn't, however, a critical analysis of the study's shortcomings at this time. For example, only two metal concentrations were examined, and the initial seedling stages under controlled settings were the primary focus. The findings' transparency and contextualization might be enhanced by including a specific paragraph that addresses these and any additional methodological or interpretive limitations.

3. Citations & Reference Style:

The manuscript use author-year citations; nevertheless, PLOS ONE criteria state that references must be numbered and enclosed in square brackets using the Vancouver style. Kindly update the reference style used in the work to conform to these guidelines.

4. Language and Structure:

A comprehensive evaluation of the manuscript would help to fix any grammar mistakes and enhance the sentence structure. If left unchecked, these problems may cause the text to lose its coherence and flow, which could impair reader understanding.

Though fixing the aforementioned issues will greatly improve the manuscript's quality and conformity to journal standards, overall, this work provides valuable information on the phytotoxicity of heavy metals in Cucurbita pepo. A revised version is something I'm eager to witness.

**Reviewer #2:**  Suggestions for Improvement

Abstract

Some sentences are long and packed with multiple results, which may overwhelm the reader. Splitting into shorter sentences will improve clarity.

You sometimes present results as % reduction (SVIW, seedling length) and sometimes as absolute or relative increases (proteins, sugars). Consider keeping the format parallel (e.g., always “increased by X%” / “decreased by Y%”).

The phrase “Cu at 100 µM transiently elevated cotyledon sugars (+63%), reflecting its dual role as a micronutrient and stressor” is good, but a bit dense. Simplify: “At 100 µM, Cu increased cotyledon sugars by 63%, suggesting its dual role...”.

The multivariate analysis (PCA) could be explained more directly. Instead of “PCA: 79.7% variance, F1”, say “Principal component analysis showed Cd explained 79.7% of the variance in metabolic disruption”.

Abstract structure: PLOS ONE prefers IMRaD (Introduction, Methods, Results, and Discussion/Conclusion) style even in abstracts. This is mostly there, but can be sharpened.

Keywords: You might add “Principal component analysis” or “Heavy metal stress” for indexing purposes.

Material and Method section

Some sentences are very long and repetitive (e.g., “Radicle, hypocotyl and total seedling length: measured…” appears twice). Merge or simplify.

Avoid redundancy—methods should be precise but concise.

Write µM consistently (avoid mix of μM and µM).

Always keep a space between number and unit (e.g., “10 ml”, not “10ml”).

Ensure all cited methods (e.g., Pereira et al. 2015; Rao and Sinha 1993; Andrade et al. 2019) are fully referenced in the bibliography.

The methods are already divided into subsections, which is good. But some subsection titles can be made clearer, e.g., instead of “Protein content of cotyledons”, write “Protein quantification in cotyledons”.

Instead of “to present means ± SE and perform LSD test”, write: “Data were expressed as means ± SE, and differences among treatments were tested using LSD at p < 0.05.”

Mention assumptions if checked (normality, homogeneity of variance).

**Do you want your identity to be public for this peer review?** For information about this choice, including consent withdrawal, please see our Privacy Policy

Reviewer #1: **Yes:** Aishwarya Singh

Reviewer #2: **Yes:** Dr. Ravikumar D. Dodiya

---

## [Author Response · Author response to Decision Letter 1]

9 Oct 2025

Dear Professor Debasis Mitra,

Ref: PONE-D-25-40701

Thank you for giving us the opportunity to revise our manuscript titled "Physiological responses of Cucurbita pepo seeds to cadmium and copper stress: Differential impacts on reserve mobilization, metabolic efficiency, and growth."

We are grateful to you and the reviewers, Dr. Aishwarya Singh and Dr. Ravikumar D. Dodiya for the constructive comments and valuable suggestions, which have significantly helped us improve the quality and clarity of our manuscript.

We have carefully addressed all the points raised during the review process. The main revisions to the manuscript are summarized below:

1. Responses to Reviewer #1's Comments:

• Clarification of the research gap: As suggested, we have explicitly stated the specific knowledge gap and the significance of our study in the Introduction section (newly added paragraph).

• Critical analysis of limitations: A new subsection titled "Study Limitations and Future Directions" has been added to the Discussion to critically address the study's limitations, such as the controlled laboratory conditions and the focus on early growth stages.

• Citation and Reference Style: The reference style has been thoroughly updated to conform to PLOS ONE's Vancouver style (numbered, in square brackets). The reference list has been checked for completeness and accuracy.

• Language and Structure: The entire manuscript has undergone comprehensive language editing to correct grammatical errors and improve sentence structure, coherence, and flow. We confirm that the manuscript is now written in standard English.

2. Responses to Reviewer #2's Comments:

• Abstract:

o Long sentences have been split for better clarity.

o The presentation of results has been made more parallel, primarily using the "% increase/decrease" format.

o The phrasing regarding PCA has been simplified as recommended.

o The IMRaD structure has been sharpened, and the suggested keywords ("Heavy metal stress," "Principal component analysis") have been added.

• Materials and Methods:

o Redundant sentences (e.g., regarding growth measurements) have been merged and simplified.

o The notation "µM" is now used consistently throughout the text.

o A space has been consistently added between numbers and units (e.g., "10 ml").

o All cited methods are now fully referenced in the bibliography.

o Subsection titles have been clarified (e.g., "Protein quantification in cotyledons").

o The statistical analysis description has been rewritten for precision and now mentions that data were tested for significance using Fisher's LSD test at p < 0.05.

3. Compliance with Journal Requirements:

• Funding Statement: As requested, all funding-related text has been removed from the Acknowledgements section. We would like to clarify that the absence of specific funding was explicitly mentioned in the original manuscript to ensure full transparency. The funding statement in the submission system remains accurate and unchanged as: "The author(s) received no specific funding for this work."

• Style Requirements: The manuscript has been formatted to meet PLOS ONE's style requirements, including the correct file naming format for the revised submission.

• Data Availability: We confirm that all data underlying the findings are fully available within the manuscript and its supporting information files.

We believe that our revisions have thoroughly addressed all the concerns raised, and the manuscript is now significantly strengthened. We are hopeful that the revised version meets the high publication standards of PLOS ONE.

Sincerely,

Dr. Smail Acila

---

## [Decision Letter · Decision Letter 1]

14 Nov 2025

Dear Dr. ACILA,

Thank you for submitting your manuscript to PLOS ONE. After careful consideration, we feel that it has merit but does not fully meet PLOS ONE’s publication criteria as it currently stands. Therefore, we invite you to submit a revised version of the manuscript that addresses the points raised during the review process.

https://journals.plos.org/plosone/s/submission-guidelines#loc-laboratory-protocols . Additionally, PLOS ONE offers an option for publishing peer-reviewed Lab Protocol articles, which describe protocols hosted on protocols.io. Read more information on sharing protocols at https://plos.org/protocols?utm_medium=editorial-email&utm_source=authorletters&utm_campaign=protocols .

We look forward to receiving your revised manuscript.

Kind regards,

Debasis Mitra

Academic Editor

PLOS ONE

Journal Requirements:

Reviewers' comments:

Reviewer's Responses to Questions

**Comments to the Author**

Reviewer #1: (No Response)

Reviewer #2: All comments have been addressed

2. Is the manuscript technically sound, and do the data support the conclusions?

Reviewer #1: Partly

Reviewer #2: Yes

3. Has the statistical analysis been performed appropriately and rigorously?

Reviewer #1: Yes

Reviewer #2: Yes

4. Have the authors made all data underlying the findings in their manuscript fully available?

Reviewer #1: Yes

Reviewer #2: Yes

5. Is the manuscript presented in an intelligible fashion and written in standard English?

Reviewer #1: Yes

Reviewer #2: Yes

Reviewer #1: Overall impression/suitability for PLOS ONE

An important and topical field of research is the effect of heavy metals on seed germination, and the dataset offered here may help us better understand how plants react in the early stages of stress. A number of significant methodological, interpretive, and reporting flaws in the manuscript's current state, however, restrict the validity and reproducibility of its findings. The paper cannot be considered for publication until these problems have been carefully addressed. The work's overall quality and clarity are further enhanced by a few minor remarks and recommendations for further analysis and experimental clarifications that are included below.

Major concerns

1. Unclear experimental unit and replication — risk of pseudoreplication

Treatments were "replicated three times" (three Petri dishes), according to the methods, and each Petri dish held fifty seeds. Many outcome measures, such as lengths and dry weights, are only published for "20 samples per treatment" or for measurements at the seedling level (e.g., TSL measured on 20 samples). It's unclear if statistical testing considered Petri dishes (n=3) as separate experimental units or individual seeds as independent replicates. This is crucial because if the Petri dish is not treated as the unit, pseudoreplication occurs and the relevance is exaggerated. Kindly:

- Give specifics on the experimental unit used for each measurement and the number of tests (e.g., n = 20 seedlings but nested inside 3 dishes per treatment; indicate if seedlings were sampled from independent dishes; n = 3 biological replicates = Petri dishes for measures pooled per dish).

- Re-analyze using mixed models or aggregate to dish-level means prior to inference if the data were examined at the seed level. Provide test statistics, precise p-values, and degrees of freedom. (Snip of the method: " Each Petri plate has fifty seeds. Every therapy was carried out three times.

2. Methodological details incomplete/ambiguous

Several methodological steps need clarification/detailing:

- Seed disinfection: "10 minutes in a 10% sodium hypochlorite (NaClO, 12°) solution." The description of the concentration is unclear (10% vs. "12°"). Would you kindly offer a supplier, catalog, or standard procedure reference along with the active chlorine concentration?

- Cold soak step: seeds are immersed in distilled water at 5°C for 30 minutes "to achieve a non-germinated initial state"—a strange and perhaps superfluous phrase; explain the reasoning and whether it has an impact on imbibition and metal uptake.

- Use of treatment: Does this imply that seedlings were re-watered every day with new metal solution since 15 mL of solution was added to each Petri dish every day? Were all exposures cumulative, if so? How frequently and to what extent was the metal concentration maintained? When interpreting exposure doses, this is important.

- Samples used in biochemical experiments must be oven-dried for 48 hours at 80°C. Some metabolites and proteins may degrade at such high temperatures; cite references to support your decision or take into account lower temperatures (such as 60°C) or freeze-drying.

- Please include justification for the extraction process (solvent polarity, potential sugar loss), yields, and recovery controls, if available. Sugar extraction involves methanol extraction, which is left at room temperature for 48 hours, followed by evaporation.

3. Missing direct measurements of metal uptake / internal concentrations

It is speculative to say that the metal content in seed tissues determines penetration and that observed effects result from differential uptake. Authors should assess the amounts of Cd and Cu in cotyledons and embryonic axis (e.g., ICP-MS or AAS) for more robust mechanistic claims. At the very least, make this constraint clear and soften assertions regarding unequal penetration.

4. Claims extend beyond the data

The text occasionally asserts mechanistic disturbances (such as the suppression of auxin pathways, proteases, and α-amylase) without quantifying these enzymes or signaling components. Reword hypothetical claims or offer evidence in the form of assays (antioxidant enzymes, ROS indicators, and enzyme activity). The discussion frequently goes beyond the variables that have been measured; please use language that is more temperate (for example, "may indicate" as opposed to "demonstrates").

Minor concerns and suggested fixes

- Concentration, relevance and ecological context

The selected concentrations (100–200 µM) ought to be examined in light of the actual rhizosphere or soil pore water concentrations for agricultural areas; are these supra-physiological or environmentally reasonable? Include a brief statement contrasting the selected dosages with the environmental ranges that have been published.

- Details of the method: calibration curves and standards

For the Bradford and sugar (DuBois) tests, as well as the wavelength blanking and controls, provide the calibration curve parameters (r2, slope, and intercept) and give the detection and quantification limits.

- Edits to the language and typography

It is necessary to carefully proofread the document for repetition and grammatical errors. In the abstract and discussion, for instance, similar lines are repeated; the language should be tightened and overstatements should be avoided.

Reviewer #2: (No Response)

**Do you want your identity to be public for this peer review?** For information about this choice, including consent withdrawal, please see our Privacy Policy

Reviewer #1: **Yes:** Aishwarya Singh

Reviewer #2: **Yes:** Dr. R. D. Dodiya, Assistant Professor, Department of Entomology, Kishorbhai Institute of Agriculture Science and Research Centre, Uka Tarsadia University, Bardoli, Surat, Gujarat, India

---

## [Author Response · Author response to Decision Letter 2]

21 Dec 2025

Please find our detailed responses to the editor and reviewers’ comments in the attached “Response to Reviewers (Round 2)” document. All comments have been carefully addressed, and the manuscript has been revised accordingly.

---

## [Decision Letter · Decision Letter 2]

13 Jan 2026

Physiological responses of Cucurbita pepo seeds to cadmium and copper stress: Differential impacts on reserve mobilization, metabolic efficiency, and growth

PONE-D-25-40701R2

Dear Dr. ACILA,

We’re pleased to inform you that your manuscript has been judged scientifically suitable for publication and will be formally accepted for publication once it meets all outstanding technical requirements.

Kind regards,

Debasis Mitra

Academic Editor

PLOS One
---

## [Editor Report · Acceptance letter]

PONE-D-25-40701R2

PLOS One

Dear Dr. Acila,

I'm pleased to inform you that your manuscript has been deemed suitable for publication in PLOS One. Congratulations! Your manuscript is now being handed over to our production team.

Kind regards,

on behalf of

Dr. Debasis Mitra

Academic Editor

PLOS One